# Pathophysiological, Molecular and Therapeutic Issues of Nonalcoholic Fatty Liver Disease: An Overview

**DOI:** 10.3390/ijms20081948

**Published:** 2019-04-20

**Authors:** Simona Marchisello, Antonino Di Pino, Roberto Scicali, Francesca Urbano, Salvatore Piro, Francesco Purrello, Agata Maria Rabuazzo

**Affiliations:** Department of Clinical and Molecular Medicine, University of Catania, 95100 Catania, Italy; simomarchi91@hotmail.it (S.M.); nino_dipino@hotmail.com (A.D.P.); robertoscicali@gmail.com (R.S.); francescaurbano@hotmail.it (F.U.); spiro@unict.it (S.P.); fpurrell@unict.it (F.P.)

**Keywords:** nonalcoholic fatty liver disease, insulin resistance, metabolic syndrome, steatosis, molecular mechanisms, pathogenesis, therapy

## Abstract

Nonalcoholic Fatty Liver Disease (NAFLD) represents the leading cause of liver disease in developed countries but its diffusion is currently also emerging in Asian countries, in South America and in other developing countries. It is progressively becoming one of the main diseases responsible for hepatic insufficiency, hepatocarcinoma and the need for orthotopic liver transplantation. NAFLD is linked with metabolic syndrome in a close and bidirectional relationship. To date, NAFLD is a diagnosis of exclusion, and liver biopsy is the gold standard for diagnosis. NAFLD pathogenesis is complex and multifactorial, mainly involving genetic, metabolic and environmental factors. New concepts are constantly arising in the literature promising new diagnostic and therapeutic tools. One of the challenges will be to better characterize not only NAFLD development but overall NAFLD progression, in order to better identify NAFLD patients at higher risk of metabolic, cardiovascular and neoplastic complications. This review analyses NAFLD epidemiology and the different prevalence of the disease in distinct groups, particularly according to sex, age, body mass index, type 2 diabetes and dyslipidemia. Furthermore, the work expands on the pathophysiology of NAFLD, examining multiple-hit pathogenesis and the role of different factors in hepatic steatosis development and progression: genetics, metabolic factors and insulin resistance, diet, adipose tissue, gut microbiota, iron deposits, bile acids and circadian clock. In conclusion, the current available therapies for NAFLD will be discussed.

## 1. Nonalcoholic Fatty Liver Disease: Definition and Diagnosis

Nonalcoholic Fatty Liver Disease (NAFLD) describes liver steatosis in the absence of secondary causes of hepatic fat accumulation such as alcohol abuse, defined as a daily alcohol consumption >20 g/day for women or >30 g/day for men [1]. Alcoholic liver disease and NAFLD are histologically similar and not discernible by mere biopsy. In everyday practice, the two conditions may often coexist; these cases warrant clinical judgement. In fact, we still do not have reliable markers to recognize the percentage of damage due to metabolic factors and due to other toxins (e.g., ethanol). In addition, in NAFLD, alcohol consumption may act as a cofactor for cirrhosis development [2].

NAFLD is a diagnosis of exclusion, and conventionally there are two possible approaches to the diagnosis: invasive and non-invasive assessment. This is worthy of interest because according to the methodology used, there is great variability in the prevalence rates reported in various studies. Liver biopsy represents the gold standard for the diagnosis, requiring the presence of steatosis affecting >5% of hepatocytes. Also, non-invasive techniques can effectively quantify the composition of fat in liver tissue. For example, sonography is an easily available tool, it is cheap, and it effectively recognizes steatosis if fatty infiltration is >10% [3]. Moreover, ultrasound-based elastography, which includes transient elastography (TE), acoustic radiation force imaging (ARFI) and shear-wave elastography, shows an accuracy of approximately 90% in detecting advanced fibrosis and cirrhosis [4]. Unenhanced Computed Tomography (CT) provides high performance in diagnosing steatosis only if fatty infiltration is >30%, mainly evaluating the attenuation value of the liver, possibly compared to the spleen; moreover, CT is a more expensive method and it carries risks related to radiation exposure [5,6,7]. Magnetic resonance spectroscopy (MRS) and quantitative fat/water selective magnetic resonance imaging (MRI) are accepted methods to identify NAFLD [1,8]. MRI and MRS have higher sensitivity and specificity than ultrasonography; however, they are more expensive and not routinely available in clinical settings [9,10]. Other studies used blood tests, for example, elevated liver enzymes (e.g., Glutamic Pyruvic Transaminase GPT) as a marker of liver steatosis, but aminotransferase levels may be normal in up to 78% of patients with NAFLD [5,6,11]. Moreover, as demonstrated in the Dionysos nutrition and liver study, NAFLD shows a similar prevalence in people with and without suspected liver disease, which means that the use of markers such as GPT for a suspected fatty liver should be discouraged because of the intrinsic risk of disease underestimation [12]. Several scores, combining different serum biomarkers and metabolic parameters, have been proposed to estimate the probability of diagnosing liver steatosis [4]. This is very useful especially in population studies, where carrying out a liver sonography for mass screening is impractical, to say the least. Among the most accepted scores there are the fatty liver index (FLI) [13], the NAFLD liver fat score [14] and the SteatoTest [15]. While the last is a commercial kit, the FLI and the NAFLD liver fat score are based on serum triglycerides (TG), gamma-glutamyl transferase, insulin, aminotransferases, waist circumference, BMI and presence of metabolic syndrome or type 2 diabetes [16]. Other scores estimate the probability of having NASH or advanced fibrosis, which is very interesting since these scores allow the physician to bypass an invasive liver biopsy. Among them, the most validated scores are the NAFLD fibrosis score [17] and the FIB-4 [18], which are based on platelet count, aminotransferases and albumin. However, it seems that the best diagnostic performance to detect advanced fibrosis is obtained combining both imaging (e.g., magnetic resonance or transient elastography) and serum biomarkers [4,19,20]. Finally, an extremely interesting new area of research is the use of blood circulating RNA sequences, especially miRNA, for the diagnosis and staging of NAFLD (“liquid biopsy”) [21]; however, a full explanation of these techniques is beyond the scope of this review.

### Differential Diagnosis

As mentioned above, NAFLD is a diagnosis of exclusion. It is mandatory to exclude excessive alcohol consumption and other causes of liver steatosis such as hepatitis C virus (HCV), which nowadays is drastically diminishing thanks to novel effective direct acting agents, autoimmune hepatitis, coeliac disease, Wilson’s disease, hemocromatosis, a/hypo-betalipoproteinemia, and other rare causes of liver steatosis (Table 1).

## 2. Epidemiologic Features and Relevance of NAFLD

NAFLD represents the leading cause of liver disease in Western countries; however, its diffusion is currently emerging also in developing countries [1,36,37]. According to a recent meta-analysis conducted by Younossi et al., the global prevalence of NAFLD in the adult population is 25% [37]. The most affected regions worldwide are South America (31%) and the Middle East (32%); also, some Asian countries show elevated prevalence rates, particularly in Japan where the prevalence exceeds 50%, it is almost 50% in Korea, followed by Singapore, India and China [38]. The United States and Europe show similar data (24%), with a reported prevalence rate in Italy of 20–25% [12,16]. As demonstrated by the Dionysos Study, fatty liver was present at sonography in 70% of a population from Northern Italy with elevated alanine transaminase levels [39,40]. On the other hand, Africa presents the lowest prevalence (14%). It is interesting to note that the incidence of NAFLD is constantly increasing, paralleling the epidemy of obesity and diabetes, though it is not clear if fatty liver is the cause or the consequence of an impaired metabolic status [41,42]. Data from National Health and Nutrition Examination Surveys report that NAFLD was responsible for chronic liver disease in 46.8% of cases between 1988 and 1994, while from 1994 to 2004, it accounted for 75.1% of cases [36]. A mathematical model has recently been used to forecast how the NAFLD disease burden will change from 2016 to 2030 in eight countries representing approximately 25% of the total global population [42]. According to this study, the NAFLD prevalence in Italy will evolve from 25.4 to 29.5%, which is the highest prevalence in the analyzed countries, with higher rates of NASH (6.3% in 2030 versus 4.4% in 2016). The increasing prevalence is observed in all the countries taken into account. The highest prevalence growth will be observed in China, as a consequence of urbanization. Finally, liver-related mortality will accordingly increase, with 10490 expected deaths for liver disease in Italy in 2030 instead of the 4870 deaths observed in 2016. This is probably the result of both population aging and “diabesity” epidemics.

NAFLD is also progressively becoming one of the main diseases responsible for hepatic insufficiency, HCC and the need for orthotopic liver transplantation (OLT) [43]. According to the American Association for the Study of Liver Disease (AASLD), Nonalcoholic Steatohepatitis (NASH) represents the second cause after HCV of HCC-related OLT [44]. Even though less data are available regarding NAFLD as a cause of liver transplantation in Europe, it is clear that NASH is emerging as the real cause of many OLTs carried out for “cryptogenic cirrhosis”, estimated to be responsible of 4% of European liver transplantation [45,46,47]. According to the 2018 Annual Report of the European Liver Transplant Registry (ELTR), NASH accounts for 1% of the OLTs carried out in Europe between 1988 and 2016, and almost all cases of OLTs for NASH have been reported in the last 15 years [48]. According to the authors of the report, NASH is going to become the leading cause of OLT within the next 10 years. Notably, coexisting metabolic risk factors in these patients considerably increase the periprocedural and postprocedural risks of surgical procedures [49], so that many patients may be excluded from the transplant waiting list because of severe comorbidities and likely poor outcome [49]. NAFLD recurrence and de novo NAFLD after OLT represent a novel challenge for hepatologists and transplant surgeons [50]. In fact, patients undergoing OLT often develop metabolic complications such as arterial hypertension, diabetes and dyslipidemia, probably also because of adverse effects of immunosuppressive drugs [51].

The prevalence of NAFLD increases with age [37], and older people seem to have a higher NASH prevalence rate and higher degree of fibrosis, evaluated by TE [52], and liver biopsy [53]. This could be explained by a lower metalloproteinases activity and consequently reduced collagenolysis [54]. Moreover, the hepatic volume appears to be reduced in the elderly, probably as a result of decreased hepatic blood flow [55]; this finding could influence the results of TE. Generally, older age is associated with increased susceptibility to oxidative stress and oxidative damage, albeit contrasting data are reported in the literature [55]. An interesting study by Collins et al. connected the enhanced fibrogenesis of the aged liver with changes in the immune system and particularly in macrophage response [56].

Children are not spared from liver steatosis. Pediatric NAFLD is a well-recognized entity that is increasingly raising concerns in the endocrine and pediatric community [57]. It has been demonstrated that, although the pathological factors are similar to the adults, patients with pediatric NAFLD exhibit some histological differences that probably warrant a different histological score [58]. Even in early onset, NAFLD seems to increase the risk of hepatic insufficiency and leads to greater mortality compared to the general population of the same age and sex [59]. As shown in a recent Swedish cohort study, an elevated body mass index (BMI) in young adults is significantly associated with an increased risk of developing severe liver disease later in life [60]. Notably, puberty is associated with insulin resistance; the reason for this phenomenon is not entirely defined, but a possible role may be played by sexual hormones. Since insulin resistance favors NAFLD development, puberty may increase the risk of fatty liver and may exacerbate NASH; however, literature data are discordant on this specific issue [57,61,62].

NAFLD is strongly associated with metabolic comorbidities such as obesity (51%), type 2 diabetes (22.5%), dyslipidemia (69%), and hypertension (39.3%) [37]. These prevalence rates increase considering patients with NASH (Table 2). On the other hand, according to a project carried out by the Italian Arteriosclerosis Society, the prevalence of ultrasound-diagnosed NAFLD in a nondiabetic population with metabolic syndrome was 78.8% [63].

As shown by Lomis et al., NAFLD prevalence increases linearly with BMI, reaching an up to 14-fold higher risk at BMIs of 37.5–40 Kg/m^2^, compared to the normal weight population [64]. As expected, the absolute risk was higher in diabetic patients, for any given BMI. Particularly, concerning NAFLD risk, the presence of diabetes in a normal weight population was equivalent to a 5–10 Kg/m^2^ increase in BMI. In fact, up to 70% of patients with diabetes mellitus show fatty liver disease [65,66] and patients with NAFLD and diabetes exhibit a more severe grade of inflammation at liver biopsies and a more rapid evolution toward HCC. Koehler et al., examining the data from the Rotterdam Study, demonstrated that the coexistence of NAFLD and diabetes was associated with a higher risk of developing liver fibrosis [37,52], and the elevated values of stiffness in these patients were not influenced by age. Moreover, not only a personal history of diabetes, but also familiarity for hyperglycemia has been associated with an increased risk of developing NASH in NAFLD patients [67]. Finally, the presence of diabetes increases mortality in NAFLD patients [68]. On the other hand, it is important to mention that the presence of NAFLD increases the risk of hyperglycemia and, in already diabetic patients, intensifies the risk of metabolic decompensation [41]. Thus, the relationship between NAFLD and insulin resistance is mutual.

Although the incidence of NAFLD directly correlates with BMI [64], it should be kept in mind that also the lean population can be affected with a reported prevalence of non-obese fatty liver disease ranging from 10 to 30% [69]. BMI and waist circumference are not perfectly reliable predictors of visceral adiposity, which is the main adipose form responsible for insulin resistance and NAFLD. In fact, it is believed that insulin resistance still plays a role in these metabolically obese but normal weight (MONW) patients [70,71,72]. Several studies have demonstrated an association between lean NAFLD and metabolic risk factors such as dyslipidemia, hyperglycemia and visceral adiposity [73,74]. Moreover, fructose consumption and a cholesterol-rich diet have been linked to non-obese NAFLD development [75,76,77]. Finally, genetic and epigenetic factors probably play a fundamental role [73,78]. Interestingly, the histologic features of non-obese NAFLD are equivalent to the obese form, and mortality is similarly increased in both groups when compared to the general population [69].

Similar to metabolic syndrome, lipid status in NAFLD includes elevated TG and low density lipoprotein cholesterol (LDL), and particularly increased small dense LDL (sdLDL), with low high-density lipoprotein cholesterol (HDL)-cholesterol [79,80,81]. This atherogenic dyslipidemia may, in part, justify the elevated cardiovascular risk of these patients. In addition, postprandial lipemia, defined as a rise in TG lipoproteins after the consumption of a meal, has been demonstrated to be higher in NAFLD patients compared to controls, with an emerging role in oxidative stress and cardiovascular disease [82,83,84]. Moreover, lipoprotein(a) increases with NAFLD severity [85]. Finally, as described below, several mutations of genes involved in lipidic metabolism have been associated with NAFLD development and severity [86,87].

It is widely accepted that NAFLD is more common in men than in women [88]. This is probably due to a protective role of estrogens in premenopausal women. It has been demonstrated, indeed, that these hormones have anti-inflammatory, antioxidant, antiapoptotic and probably antifibrotic effects, and thus hepatoprotective action [89,90,91]. In addition, estrogens favor subcutaneous fat accumulation rather than ectopic visceral fat, weakening the atherogenic effects typical of the distribution of the latter [92,93]. Remarkably, visceral adipose tissue deposits release free fatty acids directly in the portal venous flow, thus directly reaching the liver [94]. After menopause, ovarian senescence dramatically increases NAFLD risk, thus the idea of “sexual dimorphism” of NAFLD is arising to describe the increased incidence of the disease in men and postmenopausal women [91,95]. This concept is partly due to metabolic differences, in terms of insulin sensitivity, reduced lipogenesis and central-type body fat distribution [93,96,97]. Moreover, it seems that the duration of estrogen deficiency correlates with fibrosis risk in NAFLD [98].

## 3. NAFLD Natural History

NAFLD includes a wide spectrum of liver damage, extending from benign hepatic fat deposition in the absence of inflammation to liver fibrosis and hepatocarcinoma (HCC). Generally, two pathologically distinct conditions are identified: nonalcoholic fatty liver (NAFL) and NASH. While NAFL is characterized by simple liver steatosis, NASH is defined by the presence of steatosis and lobular inflammation with hepatocyte ballooning degeneration, with or without any fibrosis [8,16]. Therefore, biopsy is mandatory to distinguish the two entities [99,100]. Previous studies stated that up to one third of patients with NAFLD have the more severe form of the disease [16,101,102]. It has been shown that in almost 50% of NAFLD-related HCC, patients do not have a frank cirrhosis, even though they probably have NASH instead of mere NAFL [103]. Especially in the presence of metabolic syndrome, HCC may develop in the absence of significant liver fibrosis [104]. This problem is clinically relevant: indeed, a recent study showed that NAFLD is currently the most common cause of HCC among Medicare patients [105]; furthermore, the presence of liver fibrosis at biopsy has been shown to be the best predictor of liver-related events and mortality in NAFLD [106]. On the other hand, recent data emerging from a cross-sectional liver biopsy cohort showed that one third of patients with advanced fibrosis have no NASH; in these individuals, the severity of steatosis and genetics play a major role in determining fibrosis [107]. Moreover, it is still not clear whether NASH stems from NAFL or whether it represents a distinct entity characterized by a different prognosis. A recent retrospective study conducted by McPherson et al. demonstrates that 44% of patients with NAFL can progress to NASH during a median follow-up period of 8 years and consecutively 37% of these subjects show fibrosis progression [108].

People affected by NAFLD have an increased mortality compared to the general population, mainly due to cardiovascular deaths [8,109]. Furthermore, NASH patients exhibit greater liver-related mortality due to cirrhosis complications and HCC. Particularly, the evidence of any degree of fibrosis at liver biopsy leads to decreased survival time and greater need for liver transplantation [68].

## 4. Pathophysiological Features

The pathogenesis of NAFLD is multifactorial. Genetic factors cooperate with metabolic and environmental factors to promote the accumulation of fat in hepatocytes. In the last decade of the 20th century, the most corroborated theory was the “two hit pathogenesis”. It stated that insulin resistance leads to TG deposition in the liver, thus steatosis, rendering it more susceptible to the action of second hits, such as oxidative stress, ATP depletion and endotoxins, finally leading to inflammation, fibrosis and cancer. Nowadays, this theory has been replaced by the “multiple hit pathogenesis”. This states that multiple etiopathogenic factors act in a parallel or sequential and somehow synergic way on a genetically predisposed subject, to cause NAFLD and thus defining the spectrum of the disease phenotype (Figure 1) [110]. Particularly, some patients will develop NAFL and consequently NASH, but others will directly present inflammation and fibrosis, probably because of the influence of genetic and epigenetic factors.

The hallmark feature determining NAFLD is TG accumulation in the liver, as a result of an imbalance between fatty acid (FA) influx and FA efflux [16]. An interesting study using stable isotope labelling techniques demonstrates that 74% of hepatic TG in NAFLD derives from exogenous sources, particularly from serum non-esterified fatty acids (NEFA) and from diet, while only 26% comes from hepatic de novo lipogenesis (DNL) [112]. DNL seems to be increased up to threefold in NAFLD patients compared to controls [111] and, curiously, it fails to increase postprandially, since its activity is already elevated in the fasting state. Regarding TG balance in the liver, an impaired hepatic beta-oxidation of FA contributes to NAFLD. One of the main contributors to impaired FA utilization is malonil-CoA generated during DNL. It is indeed a steric inhibitor of mithocondrial carnitine-O-palmitoyltransferase 1 (CPT1), which is the key rate-limiting enzyme for beta-oxidation [113]. TG excretion through very-low density lipoproteins (VLDL) seems to be increased in NAFLD but it is still not sufficient to compensate for the excessive inflow in the liver [111].

As mentioned above, up to one-third of NAFLD patients develop NASH, which increases the risk of fibrosis and cancer [16,101,102]. NASH develops when physiological adaptive mechanisms of the liver are overwhelmed by the excessive influx of TG, leading to lipotoxicity, inflammation, radical oxygen species (ROS) formation and hepatocellular dysfunction [16,111,114]. TG formation is a protective mechanism for the liver. However, when an excessive amount of FA reaches the hepatocyte, the mitochondria increase FA utilization through beta-oxidation and oxidative phosphorylation; furthermore, peroxisomes and endoplasmic reticulum (RE) also contribute to FA oxidation. The drawback of this process is ROS formation that, in excessive quantity, consumes the antioxidant mechanisms of the cell and leads to protein and lipid peroxidation, DNA damage and inflammation. Interestingly, mitochondrial function decreases in more advanced stages of NASH, generating a vicious cycle [115].

Concerning HCC, the pathogenesis of a cirrhotic liver has been widely described. Chronic injury with constant regeneration stimulus, oxidative stress and consequently DNA damage, act together to favor the development of liver cancer. In the presence of NASH, several other mechanisms contribute to carcinogenesis: hyperinsulinemia and anabolic stimulus [116], the lipogenic pathway with more available sources of energy [117], lipotoxicity through oxidative and probably direct pro-oncogenic effects [118,119], low-grade inflammation and visceral obesity with altered adipokine homeostasis [86,120].

### 4.1. Genetic Factors

Interethnic variations and familial aggregation, associated with the fact that NAFLD is not always sufficiently supported by the presence of environmental and metabolic factors, convinced the scientific community that genetic factors may play a key role in liver steatosis. Genome-wide association and gene expression profiling studies have associated several genes to the development and progression of NAFLD. These sequences play a central role in different pathways: lipogenesis, FA oxidation, lipoprotein transport, glucose homeostasis, detoxification and inflammation. Since the pathogenesis of NAFLD is multifactorial, the interaction with the environment is still crucial [118,121].

Palatin-like phospholipase domain containing 3 (PNPLA3) variant allele (rs738409). In 2008, Romeo et al. conducted the first genome-wide association study of nonsynonymous sequence variations in patients with NAFLD [122]. They found a strong association between fatty liver and the PNPLA3 variant rs738409 C>G p.I148M, which causes a loss of function that is statistically more prevalent in the Hispanic population, thus justifying the major incidence of NAFLD reported in this population. Interestingly, even though the I148M variant predisposes patients to liver steatosis, it seems to not influence insulin sensitivity. The presence of this variant also implies a major risk of progression of liver damage, independently from the presence of age, gender or other metabolic risk factors such as insulin sensitivity and BMI, not only in NAFLD but also in alcoholic and viral liver disease [107,123,124,125,126]. PNPLA3 is alternatively known as calcium-independent phospholipase A2 epsilon (iPLA2-ε) or adiponutrin and it is the member of a lipid metabolizing enzyme family. It owes its name to the strong analogy of sequence with patatin, a major protein of potato tubers with nonspecific lipid acyl hydrolase activity. PNPLA3 is involved in triglyceride hydrolysis in hepatocytes and it stimulates retinyl ester release from HSC. Its levels notably increase during the postprandial phase under stimulation of insulin through SREBP-1c and ChREBP [125]. This mechanism has been proposed to explain the association between liver steatosis and PNPLA3 variants: an interaction between different allele variants of PNPLA3 (E434K/434E/148I/148M) may determine liver damage, probably influencing TG release from lipid droplets [127]. Moreover, Bruschi et al. recently demonstrated that the PNPLA3 I148M variant modulates HSC activity, leading to a pro-inflammatory and pro-fibrogenetic phenotype [128]. On the other hand, the presence of the 453I variant of PNPLA3 was associated with a lower hepatic fat content. It seems reasonable that a loss of function of this gene causes a reduction in hepatic TG outflow, favoring hepatic steatosis. 

Transmembrane 6 superfamily member 2 (*TM6SF2*) is involved in VLDL secretion [129], and a loss of function due to a single nucleotide polymorphism (rs58542926) has been linked to liver steatosis, inflammation and fibrosis [130,131,132]. It is noteworthy that this mutation seems to confer cardiovascular protection, questioning the relationship between NAFLD and cardiovascular disease. This is probably due not only to the reduction of circulating apolipoprotein B lipoproteins, but also to a complex anti-inflammatory effect [133,134].

Glucokinase regulator (GCKR) is a modulator of hepatic glucokinase, and the rs1260326-T variant leads to an uncontrolled glucose uptake by hepatocytes, with increased glycolysis, decreased beta-oxidation and hepatic lipid accumulation [135,136].

Cholesteryl ester transfer protein (CETP) is involved in reverse cholesterol transport, particularly in cholesterol esters and the TG exchange between HDL and ApoB-containing lipoproteins. It is primarily derived from Kupffer cells [137]. The variant rs1800777 of this gene has recently been associated with liver steatosis and lobular inflammation in patients with biopsy proven NAFLD [138].

Sterol regulatory element-binding protein 1c (SREBP-1c), as mentioned above, is the main inducer of hepatic DNL under insulin stimulation. Musso et al. demonstrated that a single nucleotide polymorphism of this gene (rs11868035 A/G) is associated with NAFLD development, along with insulin resistance and atherogenic dyslipidemia [139].

Membrane bound O-acyltransferase domain-containing 7 (*MBOAT7*) gene variant rs641738 C>T has been associated with liver steatosis development and severity, but not with insulin resistance [140]. A study conducted by Krawczyk et al., on the other hand, demonstrated an association between this variant and liver fibrosis but not with liver steatosis [141]. Finally, Umano et al. recently reported that the rs626283 variant may predispose obese Caucasian adolescents and children to NAFLD and insulin resistance [141]. MBOAT7 is a lysophosphatidylinositol acyltransferase involved in arachidonic acid metabolism [142].

Microsomal triglyceride transfer protein (MTTP) acts as a chaperon involved in apolipoprotein B (ApoB) lipoprotein assembly [143]. The 493 G/T polymorphism has been associated with NAFLD development and metabolic syndrome [144].

Besides the above-mentioned genes, many others have been studied, including Superoxide dismutase 2 (SOD2), Glutathione S-transferase (GST), TNF-α, Peroxisome proliferator-activated receptor α (PPARA), Apolipoprotein C-III (APOC3), and Interleukin-6 (IL6) [86].

### 4.2. Insulin Resistance and Metabolic Factors

Insulin resistance has traditionally been identified as the key pathophysiological factor in NAFLD [145]; thus, many authors consider NAFLD as the hepatic manifestation of metabolic syndrome. However, in recent years the causal role of insulin resistance in determining NAFLD has been questioned, since NAFLD often precedes and, to a larger extent, predicts the development of obesity, diabetes, dyslipidemia, arterial hypertension and cardiovascular disease [16,41,146,147,148]. In other words, NAFLD and metabolic syndrome are linked in a bidirectional way.

#### 4.2.1. Does Insulin Resistance Cause NAFLD?

Insulin signaling stimulates glucose utilization and favors lipid accumulation by acting on insulin sensitive organs such as muscle, adipose tissue and liver. If insulin resistance develops, hormone-sensitive lipase is not suppressed and consequently the adipose tissue releases a great amount of NEFA into the bloodstream, leading to ectopic deposition of fat in organs such as the liver and pancreas. Several NEFA-specific transporters have been described, not only in hepatocytes, but generally in hepatic cells, including macrophages. These transporters, e.g., fatty acid transport protein (FATP5), could be a potential target of therapy aiming at reducing hepatic NEFA intake and consequently NAFLD [149].

Besides inhibiting lipolysis, insulin also stimulates hepatic DNL mainly through the sterol regulatory element binding protein 1c (SREBP1c) and the liver X receptor (LXR)—retinoid X receptor (RXR). Due to the lack of well-known reasons, this pathway seems to be spared from insulin resistance. This insulin signaling paradox, or so-called selective insulin resistance, explains the classic triad of type 2 diabetes, namely, hyperinsulinemia, hyperglycemia and hypertriglyceridemia [150]. Interestingly, it has been demonstrated that gluconeogenesis and DNL pathways diverge distally to the insulin receptor, since insulin receptor knockout (LIRKO) mice cut down not just the formation of glucose but also triglycerides [151]. An interesting point has been offered by Hijmans et al., who, embracing the concept of liver metabolic zonation, proposed a “model of zonation of insulin sensitivity along the portocentral axis” [152]. As a matter of fact, periportal hepatocytes seem to be the main factor responsible for gluconeogenesis and beta-oxidation, while pericentral cells are mainly involved in DNL and glycolisis. Characteristically, in NAFLD, steatosis is more abundant in the pericentral zone and is rare in the periportal zone [153]. Hijmans et al. propose that insulin resistance leads to increased lypolisis in adipose tissue and thus an increased amount of NEFA arriving at the liver, specifically to periportal hepatocytes due to a “first-pass” mechanism. These cells are consequently more prone to developing insulin resistance, and this explains the impaired inhibition of gluconeogenesis in an insulin resistant organism. Following this, the pancreas further increases the secretion of insulin, which through the portal flux reaches the liver. The pericentral cells, still susceptible to insulin signaling, respond by increasing DNL. Interestingly, pediatric NAFLD may characteristically involve periportal areas, presuming a different pathophysiological mechanism of formation [154]. In addition, insulin stimulates the formation of triglycerides by activating acetyl-CoA carboxylase. Therefore, in NAFLD, DNL is still aroused by insulin signaling and enhances liver fat deposits [155].

Even hyperglycemia activates DNL, but through a different pathway that involves the carbohydrate response element binding protein (ChREBP). This is activated by an intermediate of glycolysis such as glucose 6-phosphate or fructose-2,6-biphosphate. Moreover, it should be noticed that other conditions characterized by hyperglicemia do not necessarily result in steatosis. For example, type 1 diabetes leads to little accumulation of TG in the liver, maybe due to a compensation due to enhanced beta-oxidation [156].

#### 4.2.2. Does NAFLD Cause Insulin Resistance?

The fatty acid deposits in the liver worsen hepatic insulin sensitivity, probably inducing the protein kinase C-ε (PKCε) to phosphorylate and inactivate the insulin receptor [157]. On the other hand, several studies demonstrate that metabolites such as ceramides may play a central role in mediating FA-induced insulin resistance, by acting at different levels of insulin signaling [158]. It is, however, important to recognize that even though PKCε and ceramides partly explain the bidirectional link between NAFLD and metabolic syndrome, others, such as the role of hepatokines, probably close the loop [41].

#### 4.2.3. Other Metabolic Factors: The Role of Notch and The Skeletal Muscle

Notch signaling has been shown to influence virtually every metabolic organ, thus being involved in liver steatosis and fibrosis [159]. Notch signaling is an evolutionally highly conserved mechanism composed of ligands, receptors and intracellular proteins, regulated at different levels [160]. This pathway is a key mediator of cellular proliferation, differentiation and intercellular communication, especially during embryogenesis. During adulthood, it plays a role in organ regeneration and cancer [161,162]. Bi, in a recent review, analyzed the different influence of Notch signaling in metabolic organs: liver, adipose tissue, muscle, including central nervous system and the interaction with the immune system and angiogenesis [159]. Particularly, Notch signaling endorses gluconeogenesis and DNL, having a key role in hepatic insulin resistance. Notch signaling stimulates white adipose tissue differentiation, which is the main factor responsible for energy storage during hypercaloric diet. Moreover, it maintains cellular quiescence in the central nervous system and in the muscle, preserving stem cells but limiting tissue regeneration, and it favors the differentiation of an M1 macrophage phenotype, which is the main factor responsible for systemic inflammation, thus exacerbating peripheral insulin resistance. Finally, Notch signaling has been found to regulate angiogenesis, with a possible role in favoring atherosclerosis. All these mechanisms explain the concrete role of Notch signaling in NAFLD development, which is fundamentally mediated by the promotion of an insulin resistant phenotype. Confirming this concept, several animal studies show that Notch inhibition leads to an increased insulin sensitivity and novel drugs targeting Notch signaling are currently under evaluation with extremely promising results [163,164,165,166]. However, further studies are still needed to better understand Notch signaling and to definitely confirm the efficacy and safety of novel drugs targeting this pathway.

There is increasing interest in the role of sarcopenia in liver disease. The skeletal muscle plays a central role in insulin sensitivity, since it is one of the main factors responsible for insulin-induced glucose uptake [167]. Moreover, it sequesters NEFA from the blood and therefore blunts the burden of lipids reaching the liver. It seems therefore reasonable to hypothesize a role of muscle depletion in NAFLD development [168]. Accordingly, an association has recently been found between sarcopenia, steatosis and fibrosis in NAFLD [169]. In particular, recent reports are focusing on signaling molecules and hormones called ‘myokines’. They are molecules produced by the muscle, which is, indeed, not only a mere energy consumer, but is an active endocrine organ, capable of partly contrasting the inflammatory effect of adipokines [170]. Notably, this mechanism could partly explain the beneficial role of physical exercise on metabolic diseases [170]. The most studied myokine is irisin, which seems to stimulate brown adipose tissue differentiation and thermogenesis, with a protective function from metabolic syndrome, though clinical studies have shown contrasting data regarding its role in NAFLD patients [171].

### 4.3. Diet

As mentioned above, diet is responsible for 15% of liver TG in NAFLD [112]. The prevalence of NAFLD has increased remarkably worldwide with the spread of the diet of industrialized countries. Diet composition and energy intake seem to be related to NAFLD; however, more than fat consumption, it seems that carbohydrates are the main factors responsible for NAFLD [172,173]. They provide acetyl-CoA for DNL by entering the Krebs cycle, and they supply the glycerol skeleton for TG formation via triose-phosphate.

Even though glucose is the main precursor of acetyl-CoA used for DNL, fructose is an extremely lipogenic substrate [174]. It has a selective hepatic metabolism bypassing the checkpoint of phosphofructokinase-1 and supplying acetyl-Coa by entering in glycolysis through phosphorylation mediated by hepatic fructokinase. Moreover, it provides glycerol 3-phosphate for TG synthesis and seems to increase free ROS production through nonenzymatic fructosylation [172]. In contrast to glucose, fructose does not stimulate insulin secretion and chronic exposure facilitates insulin resistance [175,176]. Moreover, chronic intake of fructose has been demonstrated to alter the “microbiota fingerprint” and damage gut tight junctions; consequently, it increases bacterial translocation, favoring systemic inflammation and NASH [177,178]. Finally, fructose consumption seems to stimulate appetite by increasing the levels of ghrelin and by acting on the central nervous system, specifically on the endocannabinoid signaling pathway [178]. For all these reasons, European guidelines recommend against the consumption of fructose-containing beverages and foods [1].

### 4.4. Overweight and Obesity

As mentioned above, obesity is strictly related to NAFLD. Adipose tissue is a very active endocrine organ that produces hormones and cytokines known as adipokines or adipocytokines. It mediates endocrine, inflammatory and immune interactions, protecting or favoring insulin resistance and liver steatosis [179].

Adiponectin is the major adipose-specific adipokine and it has powerful anti-inflammatory and insulin sensitizer effects. Several studies demonstrated an inverse correlation between adiponectin levels and hepatic steatosis, TG and LDL levels [86,180,181,182]. Leptin has protective effects on the liver since it stimulates beta-oxidation and suppresses lipogenesis, besides favoring gluconeogenesis [183]. One of the most used models of NAFLD is the leptin-deficient ob/ob mouse, which is a model of hyperphagia and severe obesity with insulin resistance. Another widely studied adipokine is resistin, which is a hormone involved in glucose and lipidic homeostasis, contributing to the development of NAFLD, insulin resistance and inflammation [120,184,185]. Apart from adipokines, adipose tissue is also responsible for the production of pro-inflammatory cytokines such as tumor necrosis factor-alpha (TNF-α), which has been demonstrated to be elevated in NASH compared to simple steatosis, and IL-6, that may favor HCC development by inhibiting apoptosis and activating pro-oncogenic patterns [86,186]. Adiposity also enhances the effect of genetic variants and possibly triggers the development of NAFLD [121].

### 4.5. Gut Microbiota

In recent years, attention has been focused on the gut microbiota (GM). It seems that a leaky mucosal barrier, especially in the presence of bacterial overgrowth, allows bacterial translocation, implicating the presence in the blood of microbial products such as lipopolysaccharide (LPS). These substances, specifically pathogen-associated molecular patterns (PAMPs), interact with receptors lying not only on inflammatory cells but also on other cell types such as hepatic stellate cells (HSC) and endothelial cells [187,188]. The liver is one of the most exposed organs to bacterial translocation because it primarily receives blood from the portal vein. One of the most studied pathways is the one activated by the Toll-like receptor 4 (TLR4), which leads to NF-κB activation and release of pro-inflammatory cytokines such as IL-1β, TNF-α, IL-6, or type I interferon [189]. Moreover, TLR can also recognize damage-associated molecular patterns (DAMPs) released by damaged cells [190], and it seems to mediate FA-induced inflammation [191]. Therefore, bacterial translocation promotes the establishment of a pro-inflammatory milieu and thus, in the presence of NAFLD, promotes NASH.

### 4.6. Iron Deposits

Iron deposits seem to play a role in NAFLD progression towards inflammation and fibrosis, while their role in steatosis development is still debated. Indeed, the IIRON2 study has recently produced new evidence regarding the relationship between hepatic iron deposits and insulin sensitivity, demonstrating that higher iron concentrations are associated with higher serum adiponectin and superior insulin sensitivity [192]. On the other hand, based on the assumption that oxidative stress is one of the key physiopathological mechanisms underlying NASH, and that iron is a highly reactive element that in excessive quantities leads to ROS formation through the Fenton reaction, it is clear how iron deposits may favor hepatic lipid peroxidation and thus membrane and DNA damage with hepatic cell apoptosis and a direct cytotoxic effect [193,194,195]. Accordingly, Maliken et al. demonstrated that when iron deposits are present in liver biopsies of NAFLD patients, higher levels of oxidative stress can be observed [196]. The authors hypothesized that an excessive oxidative burden may induce apoptosis or necrosis in the hepatic cells of these patients. Moreover, iron accumulation in liver reticuloendothelial system cells has been shown to be associated with NAFLD advanced histological features such as fibrosis, portal inflammation, cellular ballooning and defined NASH [197]; these data have also been confirmed in animal models [198]. In addition, the role of iron overload in HCC development is currently under evaluation in NAFLD, apart from the well-established role in hemochromatosis patients [118]. Moreover, further studies are warranted to understand the real prognostic value of iron deposits at liver biopsy. Interestingly, hyperferritinemia, which probably identifies NAFLD patients with major hepatic iron deposition, has also been demonstrated to be an independent predictor, not only of liver steatosis, but also of fibrosis [199,200,201]. However, it should be mentioned that clinical trials examining the potential therapeutic role of phlebotomy in NAFLD have shown contrasting results regarding any benefits in terms of reduction of liver damage [202,203].

### 4.7. Bile Acids

There is also growing interest in bile acid (BA) metabolic effects, apart from the well-known digestive ones. In the last few decades, BA receptors such as farnesoid X receptor (FXR) and Takeda G-protein-coupled receptor 5 (TGR5) have been discovered [204]. As recently described in an interesting review by Arab et al. [205], these receptors seem to play important roles in glucose metabolism, by favoring insulin sensitivity, glycogen synthesis and repressing gluconeogenesis. They also play a role in lipid metabolism, by lowering LDL-cholesterol and TG levels, and mediate anti-inflammatory effects as well as acting on Kupffer cells [206]. Bile acids are physiologically metabolized by the GM in the distal small intestine and colon. Thus, there is an interesting interplay between them. On the other hand, BA can modulate GM by influencing the energy substrate and antimicrobial response [207,208]. For all these reasons, bile acid signaling may play a role in NAFLD development and progression. Novel drugs targeting BA receptors are currently under evaluation [209].

### 4.8. Circadian Clock

Many metabolic pathways typically exhibit a circadian rhythm, principally driven by sleeping/waking and fasting/feeding cycles. Bile acid synthesis is influenced by these mechanisms too, as are the immune system and inflammatory responses [210]. All these mechanisms underlie NAFLD; it is therefore legitimate to suppose a role of the circadian clock in this disease, with the central nervous system coordinating all of these factors [211,212]. Kettner et al. recently demonstrated in animal models that chronic jet lag is sufficient to induce liver steatosis and successively HCC, probably through neuroendocrine dysfunction and pro-inflammatory effects that ultimately lead to gene deregulation [213]. Moreover, other experimental models confirmed that gene mutations involved in the circadian clock system regulation predispose to NAFLD [213,214]. However, further studies are needed to better characterize the clinical and therapeutic implications of circadian rhythm regulation.

## 5. Therapy

Today, there is no licensed pharmacotherapy for NAFLD, so the treatment is fundamentally based on lifestyle intervention and every specific drug would be off label [1]. Numerous pharmacological strategies have been tested in clinical studies for NAFLD therapy and, in accordance with the above described NAFLD physiopathology, different classes of drugs can be identified: antidiabetic drugs, antioxidants, prebiotics, drugs acting on the bile acid system and lipid-lowering therapies. It should be kept in mind that, in order to prevent cardiovascular complications in these patients, it is mandatory to treat the associated cardiovascular risk factors such as dyslipidemia, arterial hypertension and diabetes [1,215].

### 5.1. Lifestyle Intervention

The cornerstone of NAFLD treatment is lifestyle intervention [16]. Physical activity should be implemented [216,217], aiming at a minimum of 150–200 min/week of moderate intensity aerobic physical activity; a dose-effect relationship has been demonstrated, pointing out that high intensity exercise should be preferred to moderate exercise whenever possible [1]. However, the most effective strategy to reduce liver fat is dietary restriction, thus the European guidelines recommend a 7–10% weight loss (“weight normative” approach) [1,218]. In fact, as demonstrated in a recent prospective real-world study, the degree of weight loss correlates with histological improvement of all NASH-related parameters, with moderate evidence if the weight loss reaches 5% and even greater results if it exceeds 10% (90% resolution of NASH in the latter case) [219]. Similar results have been shown in randomized prospective trials comparing hypocaloric diet plus vitamin E with or without orlistat; this trial showed that a greater weight loss was associated with an increased insulin sensitivity, higher adiponectin levels and reduced liver steatosis, inflammation and ballooning, independently from the adopted pharmacological intervention [220]. A hypocaloric (120–1600 Kcal/day) low-fat or low-carbohydrate diet should be encouraged [221]. In fact, it seems that as long as weight loss is achieved, the macronutrient composition of the diet is of little importance [221,222]. With this aim, bariatric surgery is giving promising results in patients with morbid obesity, leading to a reduction of biochemical markers of NAFLD and amelioration of steatosis, hepatocyte ballooning, lobular inflammation and fibrosis [223].

On the other hand, independent of weight loss, recent evidence has highlighted the importance of having a healthy diet (“weight inclusive” approach) [224,225]. In fact, Poperzi et al. demonstrated that the isocaloric Mediterranean diet reduces liver steatosis to the same extent as a low-fat diet, even though the first one is probably more effective on cardiometabolic risk factors [226]. Particularly, saturated fatty acids should be kept below the threshold of 10% of the total fat amount and replaced with polyunsaturated fatty acids [225,227,228]. Animal-derived proteins should be reduced and the consumption of olive oil, legumes, fruits, nuts, vegetables, grains and fish should be intensified. All these indications are fundamental also in accordance with the European Society of Cardiology for cardiovascular prevention, especially considering that NAFLD patients are considered at increased cardiovascular risk [225,227].

Expanding the issue of dietary considerations, it is mandatory to reduce fructose intake (e.g., soft drinks), while there are no liver-related limitations on coffee, which has been found to be beneficial for liver steatosis [1,229]. The effect of consuming alcoholic beverages is still debated, since recent evidence suggests that even a modest consumption may enhance liver damage [2,230,231].

### 5.2. Antidiabetic Drugs

Since NAFLD is often associated with insulin resistance, even though in a complex and bidirectional relationship, drugs increasing insulin sensitivity have been proposed to treat liver steatosis and prevent diabetes in these patients. Among them, the thiazolidinediones, acting on the nuclear transcription factor peroxisome proliferator-activated receptor gamma (PPAR)-γ and thus true insulin sensitizers drugs, have demonstrated the best efficacy, improving liver histology in NASH independently from the presence of diabetes [232,233]. For all these reasons, European guidelines recommend with a B2 GRADE the use of pioglitazone in diabetic and in non-diabetic (off-label) patients [1]. The drawbacks of thiazolidinediones are weight gain, bone fractures in women, bladder cancer and congestive heart failure [234].

While the cornerstone of antidiabetic therapy in type 2 diabetes is metformin, an AMP-activated protein kinase activator, its role in reducing steatosis and inflammation in NAFLD is still debated [1]. Bugianesi et al., in a Randomized Controlled Trial (RCT) with histological outcomes, demonstrated that compared to a dietary intervention or vitamin E, metformin significantly reduced aminotransferase levels and improved liver histology in nondiabetic NAFLD patients [235]. These findings were not confirmed by subsequent RCTs [236,237]. On the other hand, metformin appears to reduce HCC risk in a dose-response pattern [41,238], probably influencing the LKB1/AMPK/mTOR axis and reducing ROS formation by acting on the mitochondrial respiratory chain [239]. In conclusion, the evidence for metformin use in NAFLD patients is still inconsistent.

Glucagon-like peptide-1 (GLP1) receptor agonists are promising therapeutic agents in NAFLD. In fact, in the LEAN (Liraglutide efficacy and action in NASH) RCT phase 2 study in overweight NASH patients, liraglutide induced NASH resolution in 39% of individuals compared to 9% of the placebo group; moreover, it significantly halted fibrosis progression [240]. The adverse events were mainly diarrhea, constipation and loss of appetite. Interestingly, the authors hypothesized that the weight reduction and the better glycemic control of the liraglutide group could also reduce cardiovascular risk in overweight patients with NASH, as already demonstrated in diabetic patients at high cardiovascular risk [241]. However, it should be noted that the hepatoprotective effect of liraglutide is probably not entirely justified by an improvement of the metabolic phenotype, but it is also due to a direct hepatic effect [242]. Moreover, other GLP1 receptor agonists have been tested in clinical studies with promising results [243]. On the contrary, sitagliptin, an oral dipeptidyl peptidase-4 (DPP4) inhibitor, in a recent RCT did not significantly reduce liver steatosis in prediabetic or diabetic patients with NAFLD [244].

Finally, the sodium/glucose cotransporter 2 (SGLT2) inhibitors are raising growing interest for the treatment of NAFLD, also considering their known beneficial effect on cardiovascular prevention in people with diabetes [245]. In the E-LIFT (Effect of empagliflozin on liver fat in patients with type 2 diabetes and nonalcoholic fatty liver disease: A randomized controlled trial) Trial, empagliflozin reduced liver fat evaluated by MRI in diabetic NAFLD patients and reduced liver transaminase levels [246]. These findings did not seem to be related to a better glycemic control or a greater weight reduction in the treatment group. Similar results on liver steatosis were also reported by ipragliflozin and luseogliflozin [247,248]; dapagliflozin, in the EFFECT-II (Effects of Omega-3 Carboxylic Acids and Dapagliflozin on Liver Fat Content in Diabetic Patients) trial reduced hepatocyte injury biomarkers and liver fat content compared to placebo when used in association with omega 3 in diabetic NAFLD patients [249]. Several other studies examined the effects of SGLT2 inhibitors on NAFLD but, to the best of the authors’ knowledge, no histological data on the effect of gliflozin on NAFLD are available yet [250]. The negative energy balance derived from glucose urinary excretion and from the increased glucagon/insulin ratio that promotes substrate utilization shift from carbohydrates to lipids may explain the beneficial role of SGLT2 inhibitors in NAFLD [249,251]; the weight loss and the increased insulin sensitivity may contribute to the SGLT2 inhibitors effect though, as mentioned above, this is still unclear [248,250]. Finally, a direct anti-inflammatory effect of gliflozins has been proposed [252].

It should be mentioned that not all antidiabetic drugs are efficacious against liver steatosis [253]; for example, insulin and insulin secretagogues seem to exacerbate steatosis and increase the risk of HCC [41]. 

### 5.3. Antioxidants

Since oxidative stress represents one of the main etiopathogenic factors causing NASH, many antioxidant agents have been tested. Among them, vitamin E, in the PIVENS (Pioglitazone versus Vitamin E versus Placebo for the Treatment of Nondiabetic Patients with Nonalcoholic Steatohepatitis) trial, significantly reduced serum transaminases and improved steatosis, inflammation and ballooning in NAFLD compared to placebo [233,254]; these results were also confirmed by a recent meta-analysis on the effect of vitamin E [255]. Despite the concerns for long-term safety, because of the possible higher risk of mortality, hemorrhagic stroke and prostate cancer, the European guidelines recommend the use of vitamin E in NAFLD (B2, GRADE system) [1]. Notably, the dietary vitamin C intake inversely correlates with the incidence of NAFLD, suggesting that dietary supplementation may have a beneficial role [256]. Analogously, zinc and selenium supplementation ameliorated NAFLD histology in animal models [257].

Promising results have been given by omega 3 polyunsaturated fatty acids (PUFAs); they probably act on SREBP-1c, ChREBP and PPARs, reducing lipogenesis and increasing beta-oxidation [258,259]. Furthermore, omega 3 PUFAs exert anti-inflammatory effects through the interaction with active metabolites involved in inflammation such as TNF-alpha [260,261]; finally, they may increase insulin sensitivity [262]. Unfortunately, several differences in the design of the different studies makes it difficult to compare the PUFA efficacy on NAFLD treatment [259,260].

Pentoxifylline, a TNF-alpha inhibitor, has improved histology in NASH patients, independently from insulin sensitivity measures [263]. Although antioxidant and anti-inflammatory effects have been hypothesized, the real mechanisms underlying these effects remain unclear. The advantages of pentoxifylline are its affordability, its good tolerability and its long-term safety.

### 5.4. Drugs acting on The Gut Microbiota

As mentioned above, the gut microbiota, through bacterial translocations and PAMPs, nurtures inflammation. Prebiotics are nondigestible substances able to promote the growth of beneficial microorganisms in the gut, while probiotics are live microorganisms that, when administered in adequate amounts, confer a health benefit to the host [264]. A recent metanalysis demonstrated that prebiotics and probiotics reduce BMI and liver enzymes, also ameliorating the lipid profile but with no effect on inflammation [265]. A RCT recently proved that probiotics reduce liver fat content, serum aminotransferase levels, TNF-alpha and interleukin 6 levels [266]. A possible explanation of the beneficial effects of these therapies is the increased production of short-chain fatty acids, the delayed macronutrient absorption, the ameliorated barrier function, a modulation of the immune system and an effect on bile acid metabolism [265]. However, further studies are needed to better define the effects and the dosage needed for prebiotics and probiotics.

### 5.5. Drugs Acting on Bile Acids System

Obeticholic acid, a potent activator of the above-mentioned FXR already registered for the treatment of primary biliary cholangitis, has been tested for use in NASH patients in the FLINT (Farnesoid X Receptor Ligand Obeticholic Acid in NASH Treatment) trial. This study has been stopped for superiority, since the drug significantly improved necro-inflammation, reducing lipogenesis, increasing VLDL clearance but also through direct anti-inflammatory effects [267,268]. In a phase 2 RCT in NAFLD diabetic patients, obeticholic acid not only reduced markers of liver inflammation and fibrosis, but it also reduced insulin resistance [269]. Two phase 3 studies evaluating the role of obeticholic acid in non-cirrhotic NASH with liver fibrosis (REGENERATE ClinicalTrials.gov: NCT02548351) and in compensated cirrhosis due to NASH (REVERSE ClinicalTrials.gov: NCT03439254) are now ongoing. The adverse events of this drug are pruritus and increased LDL cholesterol. On the contrary, ursodeoxycholic acid (UDCA), a natural bile acid, did not prove to be superior to placebo in reducing steatosis, inflammation or fibrosis [270], even though it reduced serum transaminase levels and improved liver histology when used in combination with vitamin E [271].

### 5.6. Lipid-Lowering Therapies

As clearly demonstrated by the post-hoc analysis of the GREACE (GREek Atorvastatin and Coronary-heart-disease Evaluation) study [272] and the IDEAL (Incremental Decrease in Events through Aggressive Lipid Lowering) study [273], the use of statins is safe in patients with NAFLD and they also reduce the levels of serum transaminases. Moreover, statins ameliorate liver histology and, most importantly, reduce cardiovascular morbidity and mortality in NAFLD patients [272,273,274,275]. As demonstrated in a large cross-sectional cohort of patients with suspected NASH, statin use confers a dose-dependent protection towards all NAFLD spectrum, including steatosis, NASH and fibrosis [276]. The authors suppose an anti-inflammatory and possibly anti-fibrotic effect, besides the known inhibitory effect on lipid synthesis. Finally, a reduced risk of HCC has been demonstrated [277]. Despite all these beneficial effects, unfortunately, statins are still under-prescribed in NAFLD patients, principally because of an unjustified tendency to discourage their use in cases of elevated liver enzymes [278].

Another lipid-lowering agent is ezetimibe, a Niemann-Pick C1 like 1 (NPC1L1) protein inhibitor, which has been proposed to prevent and treat NAFLD [279]. The hypothesis is that a lower cholesterol inflow to the liver reduces inflammation, insulin resistance and apoptosis in hepatocytes [280]. However, according to a recent metanalysis, despite some studies showing a benefit for serum transaminase, liver steatosis and hepatocyte ballooning, the evidence for ezetimibe prescription in NAFLD is still scarce [280].

### 5.7. Novel Classes of Drugs

Novel classes of drugs are currently under clinical trials [281]. Selonsertib (GS-4997), a selective inhibitor of apoptosis signal-regulating kinase 1 (ASK1) with anti-inflammatory and anti-fibrotic properties, demonstrated a potential role in reducing fibrosis in NASH and stage 2–3 fibrosis [282]; it is now being tested in stage 3 fibrosis and compensated cirrhotic NASH patients (STELLAR3 study ClinicalTrials.gov: NCT03053050 and STELLAR4 study ClinicalTrials.gov: NCT03053063). Analogously, cenicriviroc, an antagonist of the C-C motif chemokine receptor 2/5 (CCR2 and CCR5) that plays a role in macrophage recruitment in the liver, has been tested in a randomized phase 2b study, showing anti-inflammatory and anti-fibrotic effects in NASH patients with fibrosis [283,284]; it is now being tested in a phase 3 clinical trial (AURORA study ClinicalTrials.gov: NCT03028740). Elafibranor is a dual agonist of PPARα/δ that, in a phase 2 RCT, resolved NASH and ameliorated steatosis, ballooning, lobular inflammation and fibrosis in patients with a NAFLD fibrosis score ≥ 4 [285]. Moreover, elafibranor improved serum liver enzymes, lipid profile, markers of inflammation and, in diabetic subjects, ameliorated insulin sensitivity, demonstrating a “broad spectrum activity” rather than a simple hepatic effect [286]. The exact mechanism of action of this drug is still unclear, but PPARα and PPARδ are transcription factors involved not only in glucose and fat metabolism, but also in immune response, inflammation, apoptosis and cell proliferation [286]. A multicenter phase 3 RCT study to evaluate the efficacy and safety of elafibranor in NASH without cirrhosis is ongoing (RESOLVE-IT ClinicalTrials.gov: NCT02704403). Aramchol is a fatty acid–bile acid conjugate that inhibits hepatic Stearoyl-CoA desaturase and, in a phase 2a study, reduced liver steatosis in a dose-dependent manner [287]. In the next phase 2b study, it significantly reduced liver fat, obtained NASH resolution without worsening of fibrosis, reduced serum transaminases and improved glycemic control compared to placebo in overweight or obese NASH patients with prediabetes or diabetes [288]. Emricasan, an oral pan-caspase inhibitor modulating apoptosis, in a recent RCT significantly decreased serum transaminases and caspase activity in patients with NAFLD and elevated serum aminotransferase [289]. It is noteworthy that an antifibrotic agent, Simtuzumab, which is an anti-lysyl oxidase-like monoclonal antibody, failed to reduce fibrosis in NASH patients with advanced fibrosis in a phase 2 RCT [290]. Finally, hepatokines are currently under evaluation as a potential target for NAFLD therapy and diabetes prevention [41,291].

## 6. Conclusions and Future Perspectives

NAFLD is a well-distinguished clinical entity affecting people worldwide. Its pathogenesis is complex and multifactorial, mainly involving genetic, environmental and metabolic factors. New concepts are constantly appearing in the literature, promising new diagnostic and therapeutic tools. Further studies are needed to better characterize not only NAFLD development but overall NAFLD progression, in order to better identify NAFLD patients at higher risk of metabolic, cardiovascular and neoplastic complications.

## Figures and Tables

**Figure 1 ijms-20-01948-f001:**
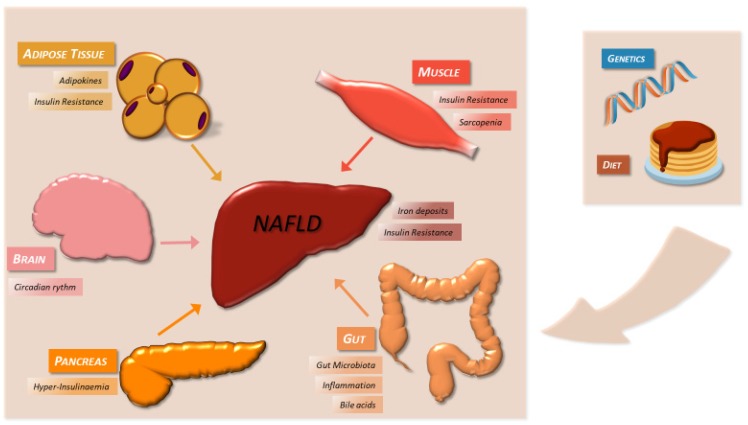
Multiple-hit pathogenesis of NAFLD [86,110,111]. Genetic factors cooperate with metabolic and environmental factors to promote the accumulation of fat in hepatocytes and successively cause inflammation, cellular death and fibrosis. Anatomically, besides the liver, the main factors are insulin sensitive organs such as adipose tissue and muscle, which respectively produce adipokines and myokines, and also promote inflammation and oxidative stress in the liver. The gut microbiota releasing PAMPs, the bile acid system and the presence of iron deposits contribute to liver damage. Finally, all these mechanisms are modulated by the brain, particularly by circadian rhythm.

**Table 1 ijms-20-01948-t001:** Differential diagnosis of NAFLD. Several conditions associated with liver steatosis are shown in this table, with possible pathogenetic mechanisms and associated references.

Conditions Associated with Liver Steatosis	Mechanism of Action	References
Alcohol (>20 g/day (women) or >30 g/day (men))	Redox state shift: fatty acid oxidation inhibition, induction of lipogenesis Altered VLDL secretion in the liver	[22]
HCV	Altered VLDL secretion in the liver Insulin resistance Mitochondrial dysfunction and oxidative stress	[23]
Medications (e.g., methotrexate, corticosteroids, valproate)	Fatty acid oxidation inhibition, induction of lipogenesis Mitochondrial dysfunction Impaired hepatic lipid secretion Insulin resistance	[24]
Lipid metabolism disorders: a/hypo-betalipoproteinaemia, Wolman’s disease	Impaired hepatic lipid secretion Impaired hydrolysis of cholesteryl esters and triglycerides	[25,26]
Metal storage disorders: Wilson’s disease	Copper-induced mitochondrial dysfunction	[27]
Autoimmune hepatitis	Drug-mediated effects	[28]
Coeliac disease	Weight gain on gluten-free diet Impaired hepatic lipid mobilization Intestinal malabsorption	[29]
Endocrine disorders: hypothyroidism, hypopituitarism, polycystic ovary syndrome	Reduced hepatic lipid utilization Insulin resistance Impaired insulin secretion	[30,31,32,33]
Starvation, parenteral nutrition	Impaired hepatic lipid secretion Reduced mitochondrial beta-oxidation	[34]
Lipodystrophy	Insulin resistance and ectopic fat accumulation	[35]

**Table 2 ijms-20-01948-t002:** Prevalence of metabolic comorbidities in patients with NAFLD and NASH—Data by Younossi [37].

	NAFLD	NASH
Obesity	51%	82%
Diabetes mellitus	23%	47%
Metabolic syndrome	41%	71%
Hyperlipidemia/dyslipidemia	69%	72%
Hypertension	39.34%	67.97%

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
