# Peer review of "Pathophysiological, Molecular and Therapeutic Issues of Nonalcoholic Fatty Liver Disease: An Overview"

_ijms, 2019, doi:10.3390/ijms20081948_

Round 1
Reviewer 1 Report
Marchisello et al. presented a timely review on 'Pathophysiological and Molecular Mechanisms in Non-alcoholic Fatty Liver Disease' that is comprehensive and important to the field. I have some minor comments those are as follows:
1. Line 10 and 61: NAFLD is not only prevalent in Western countries and developed countries, but NAFLD is also predominantly coming up in Asian countries and developing countries. For example Japan, >50%; Korea, ~50%; followed by Singapore, India, China and so on. There are enough literature to support this (Such as J Gastroenterol (2017) 52:164).
2. Line 35: Insulin resistance is only one factor, the relationship between NAFLD and metabolic syndrome is complex and bidirectionally. NAFLD is also strongly associated with abdominal obesity, hyperglycemia, hypertension, and dyslipidemia. Authors should widen the scope for this in the manuscript.
3. Table 1: Authors can elaborate the table by putting associated effects caused by medication/disease such as alcohol cause hepatic lipid accumulation, VPA caused weight gian and Ins resistance. Also, one column should be added to provide associated reference.
4. Line 174: “In the last decade of the XX century,…”. Authors may want to change it to 20thcentury.
5. Figure 1: Authors should provide a legend for this figure.
Author Response
Marchisello et al. presented a timely review on 'Pathophysiological and Molecular Mechanisms in Non-alcoholic Fatty Liver Disease' that is comprehensive and important to the field. I have some minor comments those are as follows:
Point 1: Line 10 and 61: NAFLD is not only prevalent in Western countries and developed countries, but NAFLD is also predominantly coming up in Asian countries and developing countries. For example Japan, >50%; Korea, ~50%; followed by Singapore, India, China and so on. There are enough literature to support this (Such as J Gastroenterol (2017) 52:164).
Response 1: We thank the Reviewer for this suggestion. Accordingly, we added to the manuscript the reported prevalence of NAFLD in Asian countries; we also specified that NAFLD is predominantly coming up in the developing countries. We added these considerations in the appropriate sections (line 84-89)
Point 2: Line 35: Insulin resistance is only one factor, the relationship between NAFLD and metabolic syndrome is complex and bidirectionally. NAFLD is also strongly associated with abdominal obesity, hyperglycemia, hypertension, and dyslipidemia. Authors should widen the scope for this in the manuscript.
Response 2: This is a good point. As the Reviewer suggests, the relationship between NAFLD and metabolic syndrome is complex and bidirectionally. In particular, in recent years the causal role of insulin resistance in determining NAFLD has been questioned, since NAFLD often precedes and, to a larger extent, predicts, the development of obesity, diabetes, dyslipidemia, arterial hypertension and cardiovascular disease. As suggested, this topic is now addressed in the manuscript (paragraph Insulin resistance and metabolic factors – section From NAFLD to insulin resistance – Line 336-387)
Point 3: Table 1: Authors can elaborate the table by putting associated effects caused by medication/disease such as alcohol cause hepatic lipid accumulation, VPA caused weight gian and Ins resistance. Also, one column should be added to provide associated reference.
Response 3: We thank the Reviewer for this suggestion. Accordingly, we elaborated Table 1 precising the mechanisms by which the different factors cause liver steatosis, with associated reference (Line 81-82).
Point 4: Line 174: “In the last decade of the XX century,…”. Authors may want to change it to 20thcentury.
Response 4: According with the Reviwer’s suggestion, we changed XX century in 20th century (Line 226)
Point 5: Figure 1: Authors should provide a legend for this figure.
Response 5: We thank the Reviewer for this suggestion. We have provided a legend for Figure 1 (line 237-244).

Reviewer 2 Report
GENERAL COMMENT
This is a review article on an important topic.
It is not clear to this Reviewer why principles of treatment were not addressed (I would suggest doing so).
I was also impressed by the number of sentencing referring to the epidemiology of NAFLD in the USA, which seemingly is NOT the country where these Authors work at present.
As it is, the manuscript is often redundant (see point 20 below) and there are innumerable unreferenced sentences. Bibliography is not particularly accurate, which results in wrong notions (see points 2; 6; 14).
All in all I would suggest a global reworking of the submission, without any prior commitment from the Editor’s side.
SPECIFIC COMMENT
Although both spellings are commonly used, I would suggest writing nonalcoholic (Mayo Clin Proc. 1980 ;55:434-8) rather than non-alcoholic
Throughout the manuscript (Abstract; Line 35, 95, 221, etc) “Non-Alcoholic Fatty Liver Disease (NAFLD) is considered the hepatic manifestation of metabolic syndrome” I suggest that this statement should be updated by recognizing that indeed the relationship is mutual and bi-directional (Dig Liver Dis. 2017;49:471-483; J Hepatol. 2018;68:335-352). This is much important given that 2 recent meta-analytic studies have clearly shown that the presence of NAFLD at baseline predicts incident MetS over a medium- term follow-up.
Line 31 “the two conditions could coexist” à may often
Line 34 alcohol consumption may act as a cofactor for cirrhosis development à Ref ?
Throughout the manuscript “Insulin-resistance” à insulin resistance
Line 43 “it effectively recognizes steatosis only if fatty infiltration is > 30% “ à Again, this notion is outdated and must be corrected based on novel studies (Metabolism. 2017;72:57-65).
Line 61 NAFLD represents the leading cause of liver disease in Western countriesà What about the prevalence of NAFLD in areas such as Latin America and Asia ?
62 “it is the most common cause of elevated serum aminotransferase in the United States” à I find somewhat paradoxical that European Authors should report data referring to the USA when Italy is the homeland of NAFLD epidemiology (Hepatology. 2005;42:44-52.) and the country which gave birth to transaminases (J Infect Dis. 1957 Nov-Dec;101(3):219-23. Lancet. 1972 Mar 25;1(7752):685-7). Please change this sentence accordingly.
In the section on diagnosis try to expand on “wet methods” (Castera L, et al. Gastroenterology. 2019 Jan 18. pii: S0016-5085(19)30051-4.)
“According to the American Association for the Study of Liver Disease”à Any data from Italy, Europe ? Rest of the world ?
“many patients may be excluded from the transplant waiting list because of too severe comorbidities and likely poor outcome”à Ref ?
“The incidence of NAFLD is constantly increasing, paralleling the epidemy of obesity and diabetes” à Although many authors repeat this statement, again, it fails to identify those individuals in whom NAFLD precedes the MetS
Lines 105-116 – The dangerous association of NAFLD and T2D has been incompletely summarized (Acta Diabetol. 2019;56:385-396).
“NASH is defined by the presence of liver inflammation and/or fibrosis or HCC.” This unreferenced statement is wrong and must be re-written based on standard paradigms (JAMA Intern Med. 2016;176:1083-4).
“Previous studies stated that one third of patients with NAFLD have the more severe form of
the disease” à Ref ?
Line 179 - multiple etiopathological factorsà etiopathogenic
Line 188 insulin-sensitiveà insulin sensitive
Methodically,(METHODOLOGICALLY) the role of metabolic, environmental and genetic factors should be distinguished. This interplay results in TG accumulation, inflammation and oxidative stress that ultimately lead to cellular death and fibrosis. Anatomically, besides the liver, the main factors are an insulin-sensitive organ such as adipose tissue and muscle, but also the gastrointestinal tract takes part in NAFLD development, while the nervous system is progressively assuming an important role. The hallmark feature determining NAFLD is TG accumulation in the liver, as a result of an imbalance between fatty acid (FA) influx and FA efflux. à Ref ?
As mentioned above, one-third of NAFLD patients develop NASH, which increases the risk of fibrosis and cancer. NASH develops when physiological adaptive mechanisms of the liver are overwhelmed by the excessive influx of TG, leading to lipotoxicity, inflammation, radical oxygen species (ROS) formation and hepatocellular dysfunction. à Ref ?
Line 239 “Biddinger at al., in 2008, demonstrated that” à is it important to name the Author and the year given that the relative reference allows to reader to retrieve both ?
Chapter 4.1. Metabolic syndrome and insulin-resistance has several subsections which are totally NOT linked to each other.
4.5. Genetic factors must be discussed first, not at the 5th place.
Regarding the pathogenic role of iron, if any: could these authors cite any therapeutic trials of venesection in NASH ? (e.g Hepatology. 2015;61:1555-64; World J Gastroenterol. 2014;20:3002-10). Please cite also Hepatol Commun. 2018;2:644-653.
Author Response
This is a review article on an important topic.
It is not clear to this Reviewer why principles of treatment were not addressed (I would suggest doing so).
We thank the Reviewer for this comment; therefore we added a paragraph on principles of NAFLD therapy (lines 521-712).
I was also impressed by the number of sentencing referring to the epidemiology of NAFLD in the USA, which seemingly is NOT the country where these Authors work at present.
As described more in detail below, we added the NAFLD epidemiologic data in Europe and particularly in Italy.
As it is, the manuscript is often redundant (see point 20 below) and there are innumerable unreferenced sentences. Bibliography is not particularly accurate, which results in wrong notions (see points 2; 6; 14).
Thank you for the observation, we revised the manuscript, particularly focusing on bibliography accuracy. For istance, we added the references on line 36 (Chang, Y.; Cho, Y.K.; Kim, Y.; Sung, E.; Ahn, J.; Jung, H.-S.S.; Yun, K.E.; Shin, H.; Ryu, S. Nonheavy Drinking and Worsening of Noninvasive Fibrosis Markers in Nonalcoholic Fatty Liver Disease: A Cohort Study. Hepatology 2019, 69, 64–75), on line 118 (Dare, A.J.; Plank, L.D.; Phillips, A.R.J.; Gane, E.J.; Harrison, B.; Orr, D.; Jiang, Y.; Bartlett, A.S.J.R. Additive effect of pretransplant obesity, diabetes, and cardiovascular risk factors on outcomes after liver transplantation. Liver Transplant. 2014, 20, 281–290), on line 206 (Lonardo, A.; Nascimbeni, F.; Targher, G.; Bernardi, M.; Bonino, F.; Bugianesi, E.; Casini, A.; Gastaldelli, A.; Marchesini, G.; Marra, F.; et al. AISF position paper on nonalcoholic fatty liver disease (NAFLD): Updates and future directions. Dig. Liver Dis. 2017, 49, 471–483. Williams, C.D.; Stengel, J.; Asike, M.I.; Torres, D.M.; Shaw, J.; Contreras, M.; Landt, C.L.; Harrison, S.A. Prevalence of Nonalcoholic Fatty Liver Disease and Nonalcoholic Steatohepatitis Among a Largely Middle-Aged Population Utilizing Ultrasound and Liver Biopsy: A Prospective Study. Gastroenterology 2011, 140, 124–131. Farrell, G.C.; Larter, C.Z. Nonalcoholic fatty liver disease: From steatosis to cirrhosis. Hepatology 2006, 43, 99–112) and on line 246 (Lonardo, A.; Nascimbeni, F.; Targher, G.; Bernardi, M.; Bonino, F.; Bugianesi, E.; Casini, A.; Gastaldelli, A.; Marchesini, G.; Marra, F.; et al. AISF position paper on nonalcoholic fatty liver disease (NAFLD): Updates and future directions. Dig. Liver Dis. 2017, 49, 471–483).
All in all I would suggest a global reworking of the submission, without any prior commitment from the Editor’s side.
SPECIFIC COMMENT
1. Although both spellings are commonly used, I would suggest writing nonalcoholic (Mayo Clin Proc. 1980 ;55:434-8) rather than non-alcoholic.
According with this suggestion, we modified non-alcoholic in nonalcoholic through all the manuscript.
2. Throughout the manuscript (Abstract; Line 35, 95, 221, etc) “Non-Alcoholic Fatty Liver Disease (NAFLD) is considered the hepatic manifestation of metabolic syndrome” I suggest that this statement should be updated by recognizing that indeed the relationship is mutual and bi-directional (Dig Liver Dis. 2017;49:471-483; J Hepatol. 2018;68:335-352). This is much important given that 2 recent meta-analytic studies have clearly shown that the presence of NAFLD at baseline predicts incident MetS over a medium- term follow-up.
We thank the Reviewer for outlining this important aspect. As the Reviewer suggests, the relationship between NAFLD and metabolic syndrome is complex and bidirectional. In particular, in recent years the causal role of insulin resistance in determining NAFLD has been questioned, since NAFLD often precedes and, to a larger extent, predicts the development of obesity, diabetes, dyslipidemia, arterial hypertension and cardiovascular disease. As suggested, this topic is now addressed in the manuscript (paragraph Insulin resistance and metabolic factors – section From NAFLD to insulin resistance - Lines 336-387). Thank you for the useful papers you suggested.
3. Line 31 “the two conditions could coexist” à may often.
We corrected the sentence accordingly (line 33).
4. Line 34 alcohol consumption may act as a cofactor for cirrhosis development à Ref ?
We added the reference on line 36:
Chang, Y.; Cho, Y.K.; Kim, Y.; Sung, E.; Ahn, J.; Jung, H.-S.S.; Yun, K.E.; Shin, H.; Ryu, S. Nonheavy Drinking and Worsening of Noninvasive Fibrosis Markers in Nonalcoholic Fatty Liver Disease: A Cohort Study. Hepatology 2019, 69, 64–75.
5. Throughout the manuscript “Insulin-resistance” à insulin resistance.
We substituted insulin-resistance with insulin resistance, as suggested.
6. Line 43 “it effectively recognizes steatosis only if fatty infiltration is > 30% “ à Again, this notion is outdated and must be corrected based on novel studies (Metabolism. 2017;72:57-65).
We thank the Reviewer for this comment. We updated the notion accordingly (Line 43).
7. Line 61 NAFLD represents the leading cause of liver disease in Western countriesà What about the prevalence of NAFLD in areas such as Latin America and Asia?
We thank the Reviewer for this observation. We added the reported prevalence of NAFLD in South America and in Asian countries in the appropriate sections (lines 84-89)
8. Line 62 “it is the most common cause of elevated serum aminotransferase in the United States” à I find somewhat paradoxical that European Authors should report data referring to the USA when Italy is the homeland of NAFLD epidemiology (Hepatology. 2005;42:44-52.) and the country which gave birth to transaminases (J Infect Dis. 1957 Nov-Dec;101(3):219-23. Lancet. 1972 Mar 25;1(7752):685-7). Please change this sentence accordingly.
According to the Reviewer suggestion, we inserted the data regarding European and Italian prevalence, with a referral to the Dionysos Study (lines 90-92). Moreover, a mathematical model forecasted how the NAFLD disease burden will change from 2016 to 2030 in eight countries representing approximately 25% of the total world population. According to this study, the NAFLD prevalence in Italy will evolve from 25.4% to 29.5%, that is the highest prevalence in the analysed countries, with higher rates of NASH (6.3% in 2030 versus 4.4% in 2016) (line 97-105).
Ref: Estes, C.; Anstee, Q.M.; Arias-Loste, M.T.; Bantel, H.; Bellentani, S.; Caballeria, J.; Colombo, M.; Craxi, A.; Crespo, J.; Day, C.P.; et al. Modeling NAFLD disease burden in China, France, Germany, Italy, Japan, Spain, United Kingdom, and United States for the period 2016–2030. J. Hepatol. 2018, 69, 896–904.
Thank you for the observation, which improved the quality of the paper.
9. In the section on diagnosis try to expand on “wet methods” (Castera L, et al. Gastroenterology. 2019 Jan 18. pii: S0016-5085(19)30051-4.)
We thank the Reviewer for this suggestion. We expanded the section regarding the non invasive methods available to diagnose NAFLD (lines 43-45 and lines 54-73).
10. “According to the American Association for the Study of Liver Disease”à Any data from Italy, Europe ? Rest of the world ?
According to the 2018 Annual Report of the European Liver Transplant Registry (ELTR), NASH accounts for 1% of the OLT executed in Europe between 1988 and 2016, and almost the totality of the cases has been reported in the last 15 year. According to the Authors of the report, NASH is going to become the leading cause of OLT within the next 10 years.
We added these considerations in the manuscript (lines 109-115).
11. “many patients may be excluded from the transplant waiting list because of too severe comorbidities and likely poor outcome”à Ref ?
As requested, we added the appropriate reference to the sentence (line 118):
Dare, A.J.; Plank, L.D.; Phillips, A.R.J.; Gane, E.J.; Harrison, B.; Orr, D.; Jiang, Y.; Bartlett, A.S.J.R. Additive effect of pretransplant obesity, diabetes, and cardiovascular risk factors on outcomes after liver transplantation. Liver Transplant. 2014, 20, 281–290.
12. “The incidence of NAFLD is constantly increasing, paralleling the epidemy of obesity and diabetes” à Although many authors repeat this statement, again, it fails to identify those individuals in whom NAFLD precedes the MetS.
Lines 105-116 – The dangerous association of NAFLD and T2D has been incompletely summarized (Acta Diabetol. 2019;56:385-396).
Thank you for the observations. We adjusted the statement accordingly (lines 93-95 and lines 162-165). Furthermore, we stated that the relationship between NAFLD and metabolic syndrome is complex and bidirectional in the appropriate section (paragraph Insulin resistance and metabolic factors – section From NAFLD to insulin resistance - Lines 336-387).
13. “NASH is defined by the presence of liver inflammation and/or fibrosis or HCC.” This unreferenced statement is wrong and must be re-written based on standard paradigms (JAMA Intern Med. 2016;176:1083-4).
We appreciate the comment. The definition of NASH has been updated according to the recent literature; references have been provided (lines 203-204).
14. “Previous studies stated that one third of patients with NAFLD have the more severe form of the disease” à Ref ?
Appropriate references have been provided for this sentence (line 206):
· Lonardo, A.; Nascimbeni, F.; Targher, G.; Bernardi, M.; Bonino, F.; Bugianesi, E.; Casini, A.; Gastaldelli, A.; Marchesini, G.; Marra, F.; et al. AISF position paper on nonalcoholic fatty liver disease (NAFLD): Updates and future directions. Dig. Liver Dis. 2017, 49, 471–483.
· Williams, C.D.; Stengel, J.; Asike, M.I.; Torres, D.M.; Shaw, J.; Contreras, M.; Landt, C.L.; Harrison, S.A. Prevalence of Nonalcoholic Fatty Liver Disease and Nonalcoholic Steatohepatitis Among a Largely Middle-Aged Population Utilizing Ultrasound and Liver Biopsy: A Prospective Study. Gastroenterology 2011, 140, 124–131.
· Farrell, G.C.; Larter, C.Z. Nonalcoholic fatty liver disease: From steatosis to cirrhosis. Hepatology 2006, 43, 99–112.
15. Line 179 - multiple etiopathological factorsà etiopathogenic.
We substituted etiopathological with etiopathogenic, as suggested (Line 231).
16. Line 188 insulin-sensitiveà insulin sensitive.
We substituted insulin-sensitive with insulin sensitive, as suggested, throughout the manuscript.
17. Methodically,(METHODOLOGICALLY) the role of metabolic, environmental and genetic factors should be distinguished. This interplay results in TG accumulation, inflammation and oxidative stress that ultimately lead to cellular death and fibrosis. Anatomically, besides the liver, the main factors are an insulin-sensitive organ such as adipose tissue and muscle, but also the gastrointestinal tract takes part in NAFLD development, while the nervous system is progressively assuming an important role. The hallmark feature determining NAFLD is TG accumulation in the liver, as a result of an imbalance between fatty acid (FA) influx and FA efflux. à Ref ?
In order to avoid redundancy, this paragraph has been re-elaborated (Figure 1 legend – lines 237-244): “genetic factors cooperate with metabolic and environmental factors to promote the accumulation of fat in hepatocytes and successively inflammation, cellular death and fibrosis. Anatomically, besides the liver, the main factors are insulin sensitive organs such as adipose tissue and muscle, which respectively producing adipokines and myokines, also promote inflammation and oxidative stress in the liver. The gut microbiota releasing PAMPs, the bile acids system and the presence of iron deposits contribute to liver damage. Finally, all these mechanisms are modulated by the brain, particularly by circadian rhythm”.
Appropriate reference has been provided for the last sentence (line 246):
Lonardo, A.; Nascimbeni, F.; Targher, G.; Bernardi, M.; Bonino, F.; Bugianesi, E.; Casini, A.; Gastaldelli, A.; Marchesini, G.; Marra, F.; et al. AISF position paper on nonalcoholic fatty liver disease (NAFLD): Updates and future directions. Dig. Liver Dis. 2017, 49, 471–483.
18. As mentioned above, one-third of NAFLD patients develop NASH, which increases the risk of fibrosis and cancer. NASH develops when physiological adaptive mechanisms of the liver are overwhelmed by the excessive influx of TG, leading to lipotoxicity, inflammation, radical oxygen species (ROS) formation and hepatocellular dysfunction. à Ref ?
Appropriate references have been provided for this sentences (line 257-260):
· Lonardo, A.; Nascimbeni, F.; Targher, G.; Bernardi, M.; Bonino, F.; Bugianesi, E.; Casini, A.; Gastaldelli, A.; Marchesini, G.; Marra, F.; et al. AISF position paper on nonalcoholic fatty liver disease (NAFLD): Updates and future directions. Dig. Liver Dis. 2017, 49, 471–483.
· Williams, C.D.; Stengel, J.; Asike, M.I.; Torres, D.M.; Shaw, J.; Contreras, M.; Landt, C.L.; Harrison, S.A. Prevalence of Nonalcoholic Fatty Liver Disease and Nonalcoholic Steatohepatitis Among a Largely Middle-Aged Population Utilizing Ultrasound and Liver Biopsy: A Prospective Study. Gastroenterology 2011, 140, 124–131.
· Farrell, G.C.; Larter, C.Z. Nonalcoholic fatty liver disease: From steatosis to cirrhosis. Hepatology 2006, 43, 99–112.
· Arab, J.P.; Arrese, M.; Trauner, M. Recent Insights into the Pathogenesis of Nonalcoholic Fatty Liver Disease. Annu. Rev. Pathol. Mech. Dis. 2018, 13, 321–350.
· Trauner, M.; Arrese, M.; Wagner, M. Fatty liver and lipotoxicity. Biochim. Biophys. Acta - Mol. Cell Biol. Lipids 2010, 1801, 299–310.
19. Line 239 “Biddinger at al., in 2008, demonstrated that” à is it important to name the Author and the year given that the relative reference allows to reader to retrieve both ?
The name and the year of the article has been removed (line 356-359).
20. Chapter 4.1. Metabolic syndrome and insulin-resistance has several subsections which are totally NOT linked to each other.
Thank you for the observation. The paragraph has been re-named “Insulin resistance and metabolic factors”, in order to expand the issue of the complex and mutual relationship between NAFLD and insulin resistance. Moreover, in order to analyse metabolic factors not strictly related to insulin sensitivity, a third subsection has been added. The paragraph has therefore been divided in 3 subsections (lines 336-422):
· Does insulin resistance cause NAFLD?
· Does NAFLD cause insulin resistance?
· Other metabolic factors: the role of Notch and the skeletal muscle
21. 4.5. Genetic factors must be discussed first, not at the 5th place.
Thank you for the observation. We moved the genetic factors at the first place (line 275).
22. Regarding the pathogenic role of iron, if any: could these authors cite any therapeutic trials of venesection in NASH ? (e.g Hepatology. 2015;61:1555-64; World J Gastroenterol. 2014;20:3002-10). Please cite also Hepatol Commun. 2018;2:644-653.
Thank you for the suggestion. The relationship between iron deposits and insulin sensitivity has been mentioned (lines 475-478). In addition, therapeutic trials of venesection are now cited, with the appropriate references (lines 494-496).

Reviewer 3 Report
This manuscript provides a comprehensive information on pathophysiology, epidemialogy, and diagnosis of NAFLD. There are minor issues that need to be addressed:
1- Some sentences have English errors and are not readable.
2- Figure 1 does not have a legend. Write a legend for it by explaining the figure.
3- Line 94 "Notably, puberty is associated with insulin-resistance and it therefore promotes liver steatosis", why puberty is associated with liver steatosis?
Author Response
This manuscript provides a comprehensive information on pathophysiology, epidemialogy, and diagnosis of NAFLD. There are minor issues that need to be addressed:
1. Some sentences have English errors and are not readable.
We thank the Reviewer for this comment. The language has been rechecked from the Scientific Bureau of the University of Catania for language support, as requested.
2. Figure 1 does not have a legend. Write a legend for it by explaining the figure.
This is a good point. We have provided a legend for Figure 1.
3. Line 94 "Notably, puberty is associated with insulin-resistance and it therefore promotes liver steatosis", why puberty is associated with liver steatosis?
The pubertal stage is associated to a decrease in insulin sensitivity. The reason for this phenomenon is not entirely defined, but a possible role may be played by sexual hormones. Since insulin resistance favours NAFLD development, puberty may increase the risk of fatty liver; however, literature data are discordant on this specific issue. Appropriate references are provided (line 139-143):
· Mann, J.P. Paediatric NAFLD: more than just small adults. Lancet Gastroenterol. Hepatol. 2018, 3, 222
· Kelsey, M.M.; Zeitler, P.S. Insulin Resistance of Puberty. Curr. Diab. Rep. 2016, 16, 64
· Temple, J.L.; Cordero, P.; Li, J.; Nguyen, V.; Oben, J.A. A Guide to Non-Alcoholic Fatty Liver Disease in Childhood and Adolescence. Int. J. Mol. Sci. 2016, 17.

Round 2
Reviewer 2 Report
These Authors have to be complimented for their remarkable effort in following this Reviewer's suggestion. As a result, this submission is much improved. I would recommend , however to strike the word ETIOPATHOLOGICAL (please, use ETIOPATHOGENIC instead). Moreover, please note that in English (at variance with Italian), the word EVIDENCE is uncountable. Stated otherwise, it must never be used as plural. Therefore, throughout the manuscript DELETE "EVIDENCES" and rather use EVIDENCE also to allude to multiple LINES OF EVIDENCE.